# Performance of ^18^F-FDG PET/CT in Selecting Thyroid Nodules with Indeterminate Fine-Needle Aspiration Cytology for Surgery. A Systematic Review and a Meta-Analysis

**DOI:** 10.3390/jcm8091333

**Published:** 2019-08-28

**Authors:** Marco Castellana, Pierpaolo Trimboli, Arnoldo Piccardo, Luca Giovanella, Giorgio Treglia

**Affiliations:** 1Section of Internal Medicine, Endocrinology, Andrology and Metabolic Diseases, Department of Emergency and Organ Transplantation, University of Bari Aldo Moro, IT-70124 Bari, Italy; 2Clinic for Nuclear Medicine and Competence Center for Thyroid Diseases, Imaging Institute of Southern Switzerland, Ente Ospedaliero Cantonale, CH-6500 Bellinzona, Switzerland; 3Department of Nuclear Medicine, Galliera Hospital, IT-16128 Genoa, Italy; 4Department of Nuclear Medicine, University Hospital of Zürich and University of Zürich, CH-8091 Zürich, Switzerland; 5Health Technology Assessment Unit, General Directorate, Ente Ospedaliero Cantonale, CH-6500 Bellinzona, Switzerland; 6Department of Nuclear Medicine and Molecular Imaging, Lausanne University Hospital and University of Lausanne, CH-1011 Lausanne, Switzerland

**Keywords:** thyroid nodule, PET/CT, FDG, systematic review, meta-analysis, diagnostic performance

## Abstract

Thyroid nodules with indeterminate fine-needle aspiration cytology (FNA) represent a major challenge in clinical practice. We conducted a systematic review and meta-analysis evaluating the ability of hybrid imaging using fluorine-18-fluorodeoxyglucose positron emission tomography/computed tomography (^18^F-FDG PET/CT) to appropriately select these nodules for surgery. PubMed, CENTRAL, Scopus, and Web of Science were searched until July 2019. Original articles reporting data on the performance of ^18^F-FDG PET/CT in thyroid nodules with indeterminate FNA were included. Summary operating points including 95% confidence interval values (95% CI) were estimated using a random-effects model. Out of 786 retrieved papers, eight studies evaluating 104 malignant and 327 benign thyroid nodules were included. The pooled positive and negative likelihood ratios (LR+ and LR-) and diagnostic odds ratio (DOR) of ^18^F-FDG PET/CT were 1.7 (95% CI: 1.4–2.0), 0.4 (95% CI: 0.2–0.7), and 3.5 (95% CI: 1.7–7.1), respectively. No heterogeneity was found for LR+ and DOR. In patients with thyroid nodules with indeterminate FNA, ^18^F-FDG PET/CT has a moderate ability to correctly discriminate malignant from benign lesions and could represent a reliable option to reduce unnecessary diagnostic surgeries. However, further studies using standardized criteria for interpretation are needed to confirm the reproducibility of these findings.

## 1. Introduction

Thyroid nodule is a common finding. In iodine-sufficient regions, approximately 1% of men and 5% of women present with a palpable lesion; in ultrasound (US) studies performed in the general population, its frequency increases up to 70% [1]. Even if thyroid nodules are benign in most of the cases, however, a final diagnosis can only be obtained on histology, following either lobectomy or total thyroidectomy. Then, several steps have been introduced to stratify the risk of malignancy and reduce the number of unnecessary surgical operations. These include a detailed clinical evaluation, laboratory assessment including thyroid stimulating hormone (TSH) measurement, US and fine-needle aspiration cytology (FNA) [2,3]. Particularly, cytological classification systems have been developed to improve the communication between cytopathologists and clinicians, and proved to significantly increase the proportion of malignant nodules among resected ones [4]. Four- to six-tiered reporting systems have been developed by different societies and institutions, including the Bethesda System for Reporting Thyroid Cytopathology (TBSRTC), the Italian Consensus for the Classification and Reporting of Thyroid Cytology (ICCRTC) and the Reporting of Thyroid Cytology Specimens of the Royal College of Pathologists included in the British Thyroid Association Guidelines for the Management of Thyroid Cancer (BTA) [5,6,7,8,9].

The main limit of thyroid FNA is represented by indeterminate reports, almost follicular patterned lesions, which are found in up to 25% of different institutional series. Given that only one out of four of these thyroid nodules with indeterminate FNA is anticipated as cancer, to preoperatively detect those thyroid nodules with lower risk which do not require surgery represents one major challenge in clinical practice [10]. Then, a subclassification of thyroid nodules with indeterminate FNA was proposed: atypia of undetermined significance/follicular lesion of undetermined significance (AUS/FLUS) (low-risk) and follicular neoplasm/suspicious for a follicular neoplasm (FN/SFN) (high-risk) in TBSRTC; TIR3A (low-risk) and TIR3B (high-risk) in ICCRTC; Thy3a (low-risk) and Thy3f (high-risk) in BTA. Low-risk lesions should be generally managed conservatively, while high-risk ones should be addressed to surgery [5,6,7,8,9]. The reliability of these systems has been evaluated and an actual difference between the low- and high-risk categories was found only for ICCRTC [11,12].

Other options for differentiating malignant and benign thyroid nodules with indeterminate FNA include US, ^99m^Tc-MIBI thyroid scintigraphy, immunocytochemistry and molecular testing [13,14,15,16,17,18,19]. The results of the first two imaging modalities are promising, but no definitive evidence has been obtained. Immunocytochemistry is relatively inexpensive, widely available and with a moderate diagnostic accuracy, but still limited to justify surgical decision making. Molecular methods with the highest performance are restricted to few referral laboratories and still expensive. Then, no consensus on the best strategy to adopt in this setting has been reached so far [13,14,15,16,17,18,19,20].

Among all imaging modalities, positron emission tomography/computed tomography using the glucose analogue fluorine-18 fluorodeoxyglucose (^18^F-FDG PET/CT) can hold a role in this context [19,20,21,22,23,24]. The rationale for using this method in the characterization of thyroid nodules is that many thyroid malignancies usually metabolize glucose at a higher rate compared to normal thyroid tissue or benign tumors [19,20,21,22,23,24]. Furthermore, the combination of both functional and anatomic imaging modalities by using PET/CT tomographs may improve diagnostic accuracy and localization of thyroid malignancies compared to PET only [19,20,21,22,23,24]. According to current guidelines, ^18^F-FDG PET is not routinely recommended for the assessment of thyroid nodules with indeterminate FNA, but no high-level evidence is reported for ^18^F-FDG PET/CT [3]. Several papers attempted to evaluate the diagnostic performance of ^18^F-FDG PET or PET/CT in evaluating thyroid nodules with indeterminate FNA and heterogeneous results were found by previous meta-analyses on this topic published in 2011 and 2013 [25,26], thus limiting applicability of findings of these studies to clinical practice.

We planned an updated systematic review and meta-analysis of studies reporting the performance of ^18^F-FDG PET/CT in selecting thyroid nodules with indeterminate FNA for surgery, excluding studies performing ^18^F-FDG PET only, because hybrid ^18^F-FDG PET/CT is the current state of the art [27]. 

## 2. Methods

The systematic review and the meta-analysis were performed in accordance with the Preferred Reporting Items for a Systematic Review and Meta-analysis of Diagnostic Test Accuracy Studies (PRISMA-DTA) [28].

### 2.1. Search Strategy

A six-step search strategy was planned. As first step, sentinel studies were searched in PubMed. As second step, keywords and MeSH terms were identified in PubMed. Thirdly, in order to test the strategy, the terms “PET/CT” and “indeterminate thyroid” (including TIR3, TIR3A, TIR3B, Thy3a, Thy3f, AUS/FLUS, FN/SFN) were searched in PubMed (search string reported in Appendix B). Fourthly, PubMed, CENTRAL, Scopus and Web of Science were searched. Fifthly, studies reporting the diagnostic performance of ^18^F-FDG PET/CT in thyroid nodules with indeterminate FNA were included. Studies focusing on pediatric patients, specific subgroups of thyroid nodules (i.e., incidental thyroid uptake) or nodules with inconclusive or suspicious of malignancy FNA were excluded. Finally, references of included studies were screened for additional papers. The last search was performed on 9 July 2019. No language neither time restriction was adopted. Two investigators (M.C. and G.T.) independently and in duplicate searched papers, screened titles and abstracts of the retrieved articles, reviewed the full-texts and selected articles for their inclusion.

### 2.2. Data Extraction

The following information was extracted independently and in duplicate by two investigators (M.C. and G.T.) in a piloted form: (1) general information on the study (author, year of publication, country, study type, study period, population, number of nodules, final diagnosis); (2) index test; (3) reference standard; (4) number of nodules classified as true negative, true positive, false negative, false positive. For example, a thyroid nodule was classified as true positive if a focal uptake was found on ^18^F-FDG PET/CT in a malignant thyroid nodule. Diffuse ^18^F-FDG uptake in the thyroid gland or examinations with uncertain findings were considered as negative. Data were cross-checked and any discrepancy was discussed. Authors of original articles were contacted in case of missing information.

### 2.3. Quality Assessment

The risk of bias of included studies was assessed independently by two reviewers (M.C. and G.T.) through the Quality Assessment of Diagnostic Accuracy Studies (QUADAS-2) tool for the following aspects: patient selection; index test; reference standard; flow and timing. Risk of bias and concerns about applicability were rated as low, high or unclear [29]. Data presentation was arranged using RevMan 5.3 (the Cochrane Collaboration, 2014, Copenhagen, Denmark).

### 2.4. Data Analysis

The characteristics of included studies were summarized. Then, a diagnostic performance meta-analysis on ^18^F-FDG PET/CT in selecting thyroid nodules with indeterminate FNA for surgery was carried out. The primary outcome was the diagnostic odds ratio (DOR) of ^18^F-FDG PET/CT. The secondary outcomes were: sensitivity, specificity, positive predictive value (PPV), negative predictive value (NPV), positive and negative likelihood ratio (LR+ and LR-). We plotted estimates of sensitivity and specificity on coupled forest plots. Summary operating points including sensitivity, specificity, NPV, PPV, LR+, LR-, and DOR, with 95% confidence interval (95%), were estimated [30,31]. DOR provides a single measure of test performance and it does not depend on the prevalence of the disease that the test is used for [32]; it is equal to LR+/LR- and, in our specific study, it corresponds to the odds of the surgery being indicated in a malignant nodule compared with the odds of the surgery being indicated in a benign one. The value of a DOR ranges from 0 to infinity, with higher values indicating better discriminatory test performance. A value of 1 means that a test does not discriminate between patients with benign and malignant thyroid nodules. Values lower than 1 point to improper test interpretation; a value greater than 1 indicates discriminatory capacity, which will be greater the greater the value [32]. 

A bivariate random-effects model was used for the pooled analysis of sensitivity and specificity; a random-effects model was used for the pooled analysis of the remaining metrics [30,33]. All analyses were performed on a per lesion basis and carried out using OpenMeta[Analyst] (Rockville, Maryland, United States). Heterogeneity between studies was assessed by using I^2^, with 50% or higher values regarded as high heterogeneity. Publication bias was not evaluated, because of uncertainty about the determinants for diagnostic accuracy studies and the inadequacy of tests for detecting funnel plot asymmetry [33]. A *p* < 0.05 was regarded as significant.

## 3. Results

### 3.1. Literature Search

A total of 786 papers were found (50 on PubMed, 669 on Scopus, 58 on Web of Science and 9 on CENTRAL). After removal of 113 duplicates, 673 articles were analyzed for title and abstract; 634 records were excluded (guidelines, reviews, meta-analyses, case reports, imaging modalities other than PET/CT (e.g., elastography), tracers other than FDG (e.g., choline), thyroid disease other than thyroid nodule with indeterminate FNA (e.g., thyroid carcinoma, non-diagnostic cytology), not within the field of the review, studies not in humans. The remaining 39 papers were retrieved in full-text and 8 articles were finally included in the systematic review (Figure 1) [34,35,36,37,38,39,40,41]. No article was added after screening the references of these papers.

### 3.2. Qualitative Analysis (Systematic Review)

The characteristics of the included articles are summarized in Table 1. The papers were published between 2005 and 2018, had sample sizes ranging from 16 to 108 thyroid nodules with indeterminate FNA. Participants were adult outpatients who had been diagnosed with a thyroid nodule with indeterminate FNA and were known to be undergoing surgery, underwent ^18^F-FDG PET/CT and had a histological diagnosis. Five studies were prospective, and two retrospective cohorts, whereas the design was not clearly stated in one paper. Three studies were carried in the USA, one in Canada, one in Denmark, one in France, one in Spain, and one in Italy and Switzerland.The ^18^F-FDG PET/CT protocol is reported in Table 2. The reference standard for malignant and benign diagnosis was represented by histology. The prevalence of malignancy ranged from 4% to 50%. Overall, 104 malignant and 327 benign nodules were included in the present review (Appendix A).

### 3.3. Quality Assessment

The risk of bias of the included studies is shown in the Appendix A. Overall, we found a low risk of bias: in most studies consecutive patients with thyroid nodules with indeterminate FNA were included in a specific period; ^18^F-FDG PET/CT was evaluated before the final diagnosis or, in retrospective studies, researchers were blinded to final diagnosis [36,38]. We rated reference standard bias as high since histology was commonly performed in the knowledge of the index test. We rated flow and timing bias as low since thyroid nodule is a chronic condition. The only exceptions to the statements above included three studies in which patient selection risk of bias was rated as unclear since no information on a consecutive or random enrollment was reported [35,38,40]. Three studies excluded nodules depending on size or TSH levels, thus patient selection applicability concerns item was rated as high. Particularly, in one study only nodules larger than 10 mm and with a TSH 0.5–4 mUI L^−1^ were included; in another one, only nodules larger than 5 mm were included; in the last one, only nodules larger than 10 mm and with a TSH 1–4 mUI L^−1^ were included [36,39,40]. Finally, since we used histology as reference standard, it implies that only surgically removed thyroid nodules were included, possibly leading to a further selection bias.

### 3.4. Quantitative Analysis (Meta-Analysis)

The forest plots of sensitivity and specificity of ^18^F-FDG PET/CT in each study in selecting thyroid nodules with indeterminate FNA for surgery is shown in Figure 2. The pooled sensitivity, specificity, PPV, and NPV were 74%, 58%, 34%, and 74%, respectively. Since these summary operating points are influenced by the prevalence of the disease in the population tested, we estimated the pooled LR+, LR-, and DOR, which are independent of disease prevalence and thus characteristics of ^18^F-FDG PET/CT. The pooled LR+, LR-, and DOR of ^18^F-FDG PET/CT were 1.7, 0.4, and 3.5, respectively. No significant statistical heterogeneity was found for LR+ and DOR according to the I^2^ test result (Figure 3, Table 3). Overall, the most important finding is the moderate capacity of this method in discriminating between malignant and benign thyroid nodules with indeterminate FNA, according to the pooled DOR value.

## 4. Discussion

The aim of this systematic review and meta-analysis was to identify the best available evidence on the diagnostic performance of ^18^F-FDG PET/CT in the indication of surgery in thyroid nodules with indeterminate FNA. To our knowledge, this is the first high-level evidence manuscript specifically focused on ^18^F-FDG PET/CT as previous meta-analyses pooled together data on ^18^F-FDG PET and PET/CT [25,26]. An extensive database search was performed without time or language restrictions and inclusion criteria were defined a priori. Eight studies were found, evaluating 104 malignant and 327 benign thyroid nodules. Overall, ^18^F-FDG PET/CT was found to correctly discriminate these two entities and could thus represent a reliable option to reduce unnecessary diagnostic surgeries.

Particularly, ^18^F-FDG PET/CT was characterized by moderate sensitivity and NPV, and low specificity and PPV. Therefore, a low number of false negative findings with a not-negligible rate of false positive ones should be expected. The formers were due to small-sized thyroid nodules or incidental thyroid carcinomas. It is common knowledge that the performance of ^18^F-FDG PET/CT is influenced by the size of the target lesion [34,37,41]. Also, microcarcinomas incidentally identified postoperatively, especially if distant from the assessed thyroid nodule, should have been excluded from data analysis. While some studies clearly reported this criterion [37,38], carcinomas as little as 1 mm with negative ^18^F-FDG PET/CT were classified as false negative findings in at least one paper [35]. Better diagnostic performance values of ^18^F-FDG PET/CT are expected in patients with thyroid nodules with a diameter > 1 cm [37,40]; unfortunately, we were not able to perform a subgroup analysis including only patients with thyroid nodules with diameter > 1 cm due to limited data. False positive findings at ^18^F-FDG PET/CT were due to nodule composition, with a higher standardized uptake value frequently reported for Hürthle cell adenomas according to the abundant intra-cytoplasmic mitochondria [34,36,39].

According to the calculated pooled DOR, we demonstrated that ^18^F-FDG PET/CT has a moderate ability in discriminating among malignant and benign thyroid nodules with indeterminate FNA. This finding should be considered significant in the current scenario of absence of diagnostic modalities with excellent discriminatory capacity in this setting [20]. Then, in a patient with a thyroid nodule with indeterminate FNA and a negative ^18^F-FDG PET/CT, a conservative approach based on clinical and US follow-up can be considered. On the other hand, in those patients with a with a thyroid nodule with indeterminate FNA and a positive ^18^F-FDG PET/CT, no additional information can be obtained, and patient should be managed according to current guidelines [3,42].

Several criteria for the interpretation of ^18^F-FDG PET/CT scans were proposed in the included studies. One study specifically assessed the performance of different ^18^F-FDG uptake patterns at visual analysis: a not-statistically significant higher odds ratio of malignancy for the focal and multifocal uptake patterns compared to the non-uptake pattern was found, while no differences between diffuse and non-uptake patterns was reported [41]. For the purpose of the present review, we have decided to adopt the focal ^18^F-FDG uptake pattern as positive, in line with the majority of studies. A further aspect is represented by the semi-quantitative analysis through the use of a SUV_max_ cut-off to discriminate malignant and benign lesions. A pre-specified SUV_max_ cut-off value was reported in two studies, while it was calculated with a receiver-operating characteristic curve in three; overall it ranged from 2 to 5 (Table 2). No reference value was introduced in other studies, but a lesion-based SUV_max_ calculation was performed just to support visual interpretation. Given differences in protocols and methods by which SUV_max_ are calculated from each PET/CT scanner, the proposal of a center-specific SUV_max_ cut-off seems to be reasonable [34].

Beyond the diagnostic performance, a relevant issue is represented by the costs of ^18^F-FDG PET/CT. As stated, the prevalence of thyroid nodules with indeterminate FNA is not negligible and these can be addressed to diagnostic surgery according to current guidelines [3]. In a patient with an indeterminate solitary nodule, the recommended first surgical approach is represented by thyroid lobectomy. On the other hand, total thyroidectomy may be chosen in a patient with specific characteristics (e.g., suspicious or large (>4 cm) nodule) [3]. A crucial aim of our era is to reduce invasiveness in patients with benign thyroid nodules. Indeed, thyroid surgery is available in specialized centers, however it is still associated to a 2 to 10% frequency of complications, including laryngeal nerve injury, hypoparathyroidism, hypothyroidism, neck scarring, and the known risks associated with general anesthesia [43]. Also, surgery is characterized by high costs and may not be performed in surgically high-risk patients [43]. A recent analysis demonstrated that the use of ^18^F-FDG PET/CT in selecting thyroid nodules with indeterminate FNA for surgery is cost-effective [44]. Vriens et al. reported that full implementation of preoperative ^18^F-FDG PET/CT for selecting thyroid nodules with indeterminate FNA for surgery could prevent up to 47% of current unnecessary surgery, leading to lower costs and a modest increase of health-related quality of life. Compared with an approach with diagnostic surgery in all patients and both molecular tests, ^18^F-FDG PET/CT is the least expensive alternative with similar effectiveness as the gene-expression classifier [44].

This systematic review has several limitations. The first limitation relates to the characteristics of included studies: even if a prospective design was adopted in most of them, the studies were usually small sized. Also, a wide range both in the prevalence of malignant lesions as well as in their absolute number was found, with one study including only one carcinoma [34]. In some studies, reporting was unclear or incomplete, and this was a second limitation [35,41]. Thirdly, only one study specifically excluded autonomous functioning thyroid nodules (AFTN) [37]. AFTN accounts for 5 to 10% of palpable nodules while is diagnosed in up to 20% of patients from iodine deficient regions [45,46]. Thyroid scintigraphy with either I-123-natrium iodide (^123^I) or ^99m^Tc-pertechnetate is the only method to diagnose AFTN and is recommended by ATA guidelines in patients with low serum TSH [3]. The relationship between TSH levels and thyroid autonomy, however, is largely influenced by iodine intake and normal TSH levels are detected in 50% of patients carrying AFTN [47]. Given the negligible prevalence of malignancy among AFTN and the significant number of indeterminate cytological reports that might be expected in these cases, whether different results would have been obtained if these nodules were routinely searched and excluded is unclear [3,42]. Lastly, there were not enough studies to perform a subgroup analysis for diagnostic performance of ^18^F-FDG-PET/CT in low- versus high-risk indeterminate lesions; furthermore, there were insufficient data to perform a subgroup analysis excluding thyroid nodules with diameter <1 cm. These issues should be addressed by further studies.

## 5. Conclusions

The need of stratifying the risk of thyroid nodules with indeterminate FNA is recognized and several options are available in the literature. However, poor data are reported on the diagnostic performance of ^18^F-FDG-PET/CT in selecting this specific subgroup of thyroid nodules for surgery. The main finding of this study was that ^18^F-FDG PET/CT proved to have a moderate ability to correctly discriminate between malignant and benign thyroid nodules. Further prospective studies assessing its performance using standardized criteria for interpretation and in subgroups of thyroid nodules (at low and high risk or grouped based on lesion size) are needed. Also, the present paper calls for studies comparing the diagnostic performance of all available strategies in the assessment of thyroid nodules with indeterminate FNA and ^18^F-FDG PET/CT should be included.

## Figures and Tables

**Figure 1 jcm-08-01333-f001:**
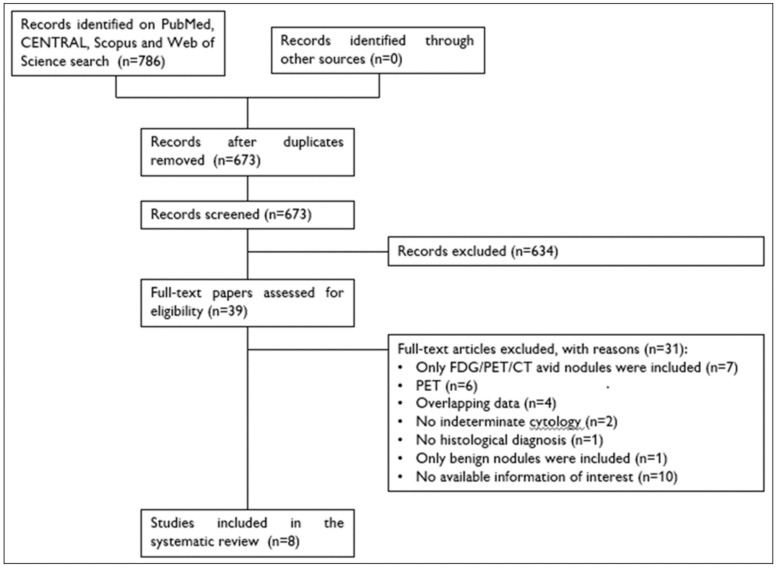
Flow chart of the search for eligible studies on the diagnostic performance of ^18^F-FDG PET/CT in selecting thyroid nodules with indeterminate FNA for surgery.

**Figure 2 jcm-08-01333-f002:**
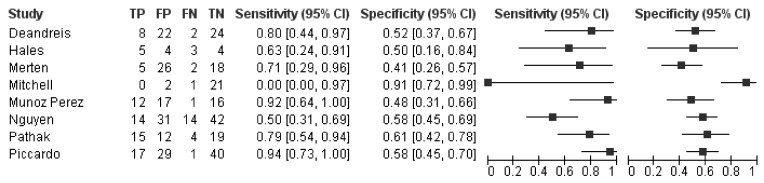
Forest plots of sensitivity and specificity of ^18^F-FDG PET/CT in selecting thyroid nodules with indeterminate FNA for surgery. Legend: TP = true positive; FP = false positive; FN = false negative; TN = true negative. References: Deandreis [36], Hales [35], Merten [38], Mitchell [34], Muñoz Pérez [37], Nguyen [41], Pathak [39], Piccardo [40].

**Figure 3 jcm-08-01333-f003:**
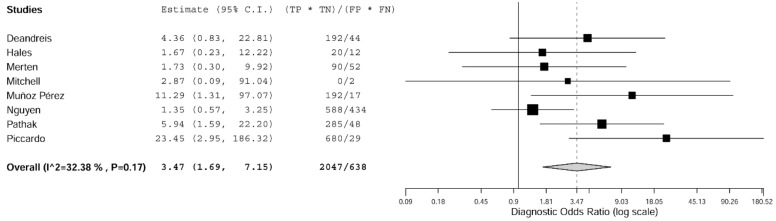
Forest plot of diagnostic odds ratio (DOR) of ^18^F-FDG PET/CT in selecting thyroid nodules with indeterminate FNA for surgery. A DOR higher than 1 indicates that the index test has discriminatory capacity among benign and malignant thyroid nodules with indeterminate FNA. Legend: TP = true positive; FP = false positive; FN = false negative; TN = true negative. References: Deandreis [36], Hales [35], Merten [38], Mitchell [34], Muñoz Pérez [37], Nguyen [41], Pathak [39], Piccardo [40].

**Table 1 jcm-08-01333-t001:** Characteristics of included studies.

First Author [ref]	Country	Study Design	Thyroid Nodules (n)	Selection Criteria
Mitchell [34]	USA	NR	24	Microfollicular pattern
Hales [35]	USA	PCS	16	Follicular or Hürthle cell lesion
Deandreis [36]	France	RCS	56	Indeterminate cytology or AUS/FLUS and FN/SFN, > 10 mm, TSH 0.5–4 mUI L^−1^
Muñoz Pérez [37]	Spain	PCS	46	Follicular or Hürthle cell neoplasm, euthyroid. Exclusion criteria: autonomous nodules, contraindication for ^18^F-FDG PET/CT or FNA, and the history or presence of another extrathyroidal cancer
Merten [38]	USA	RCS	51	Suspicious for Hürthle cell neoplasm or follicular neoplasm
Pathak [39]	Canada	PCS	50	Follicular or Hürthle cell neoplasms, > 5 mm
Piccardo [40]	Italy, Switzerland	PCS	87	Indeterminate cytology, > 10 mm, TSH 1–4 mUI L^−1^, undetectable thyroperoxidase and thyroglobulin autoantibodies
Nguyen [41]	Denmark	PCS	108	AUS/FLUS or FN/SFN. Exclusion criteria: (a) B symptoms (e.g., weight, night sweats); (b) suspicious ultrasound examination (lymph node metastasis, suspicious thyroid tumor); (c) suspicious clinical examinations (e.g., recurrent laryngeal nerve palsy); (d) lymph node metastasis and/or distant metastasis on ^18^F-FDG PET/CT, histology other than of thyroid origin

Legend: AUS/FLUS = atypia of undetermined significance/follicular lesion of undetermined significance; ^18^F-FDG PET/CT = fluorine-18 fluorodeoxyglucose positron emission tomography/computed tomography; FNA = fine-needle aspiration cytology; FN/SFN = follicular neoplasm/suspicious for a follicular neoplasm; NR = not reported; PCS = prospective cohort study; RCS = retrospective cohort study.

**Table 2 jcm-08-01333-t002:** ^18^F-FDG PET/CT protocol.

First Author, [ref]	PET/CT Scanner	^18^F-FDG Injected Activity (MBq)	Time between ^18^F-FDG Injection and Image Acquisitions (min)	SUV_max_ Cut-Off Value for Benign and Malignant Lesions
Mitchell [34]	Discovery LS (General Electric Healthcare, Chalfont St. Giles, UK)	740	60	5
Hales [35]	REVEAL XVI HiREZ (Cti Molecular Imaging, Knoxville, Tennessee, USA)	444–518	60	2
Deandreis [36]	Biograph (Siemens, Knoxville, Tennessee, USA)	118–189	60	NR
Muñoz Pérez [37]	Biograph 16 (Siemens, Knoxville, Tennessee, USA)	210–370	45–60	4.2
Merten [38]	Discovery (General Electric Healthcare, Chalfont St. Giles, UK)	444–519	60–70	5
Pathak [39]	Biograph-16 (Siemens, Malvern, Pennsylvania, USA)	185	NR	3.25
Piccardo [40]	Discovery LS (General Electric Medical Systems, Milwaukee, Wisconsin, USA) Biograph 16 (Siemens, Knoxville, Tennessee, USA)	111	50	NR
Nguyen [41]	Discovery 690 (General Electric Healthcare, Waukesha, Wisconsin, USA) Discovery VCT (General Electric Medical Systems, Milwaukee, Wisconsin, USA) Discovery RX (General Electric Medical Systems, Milwaukee, Wisconsin, USA) Discovery STE (General Electric Medical Systems, Milwaukee, Wisconsin, USA)	200–400	60	NR

Legend: ^18^F-FDG = fluorine-18 fluorodeoxyglucose; NR = not reported; SUV_max_ = maximum standardized uptake value; MBq = MegaBecquerel; min = minutes.

**Table 3 jcm-08-01333-t003:** Summary estimates of the diagnostic performance of ^18^F-FDG PET/CT in selecting thyroid nodules with indeterminate FNA for surgery.

Sensitivity (95% CI)	Specificity (95% CI)	Positive Predictive Value (95% CI)	Negative Predictive Value (95% CI)	Likelihood Ratio for Positive Results (95% CI)	Likelihood Ratio for Negative Results (95% CI)	Diagnostic Odds Ratio (95% CI)
74% (55–87)	58% (48–67)	34% (25–44)	74% (41–100)	1.7 (1.4–2.0)	0.4 (0.2–0.7)	3.5 (1.7–7.1)
I^2^ = NA	I^2^ = NA	I^2^ = 57%	I^2^ = 99%	I^2^ = 17%	I^2^ = 93%	I^2^ = 32%

Legend: I^2^ = I-square test result for heterogeneity; NA = not available for bivariate meta-analysis; 95% CI = 95% confidence interval.

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
