# Peer review of "Performance of 18F-FDG PET/CT in Selecting Thyroid Nodules with Indeterminate Fine-Needle Aspiration Cytology for Surgery. A Systematic Review and a Meta-Analysis"

_jcm, 2019, doi:10.3390/jcm8091333_

Round 1
Reviewer 1 Report
I read with interest this systematic review and meta-analysis by Castellana et al regarding the performance of 18F‐FDG PET/CT in selecting thyroid nodules with indeterminate fine‐needle aspiration cytology for surgery. The authors followed the PRISMA guidelines and overall methodology is good. However, there are some points of concern.
For the studies with missing information, PRISMA guidelines advocate authors' contact to request for the data. No effors were described for authors' contact. The authors mention the limitation of FDG PET/CT with nodules <1 cm. Can they perform a sensitivity analysis excluding studies/subgroups of nodules less than 1 cm? The description of LRs and interpretation under Data analysis is not done well. The authors conclusions are not supported by the Results. LRs
quantify the ability of the test result to modify the pre-test
probability; a LR of 1 does not change it; 0.2 or 5 produce
moderate changes; 0.1 or 10 produce large changes to the
pre-test probability and are often seen with rule-in or ruleout
test results. The LR+ was just 1.7 and LR- 0.4. The last paragraph of Introduction belongs to Methods. Line 119: histology is the reference standard, not the index test
Author Response
Reviewer's comment: For the studies with missing information, PRISMA guidelines advocate authors' contact to request for the data. No efforts were described for authors' contact.
Reply: We added this statement in the methods section of the revised manuscript "Authors of original articles were contacted in case of missing information."
Reviewer's comment: The authors mention the limitation of FDG PET/CT with nodules <1 cm. Can they perform a sensitivity analysis excluding studies/subgroups of nodules less than 1 cm?
Reply: unfortunately we have insufficient data to perform a subgroup analysis excluding studies/subgroups of nodules less than 1 cm. This was added as a limitation of our analysis. We have added this statement in the discussion: “Better diagnostic performance values of 18F-FDG PET/CT are expected in patients with thyroid nodules with a diameter > 1 cm [37,40]; unfortunately, we were not able to perform a subgroup analysis including only patients with thyroid nodules with diameter > 1 cm due to limited data.”
Reviewer's comment: The description of LRs and interpretation under Data analysis is not done well. The authors conclusions are not supported by the Results. LRs quantify the ability of the test result to modify the pre-test probability; a LR of 1 does not change it; 0.2 or 5 produce moderate changes; 0.1 or 10 produce large changes to the pre-test probability and are often seen with rule-in or ruleout test results. The LR+ was just 1.7 and LR- 0.4.
Reply: We have deleted the description and interpretation of LRs under data analysis in the revised manuscript. Furthermore we have deleted the Figure on LRs and comments on LRs in the discussion of the revised manuscript. We have focused the discussion on our primary outcome (DOR).
Reviewer's comment: the last paragraph of Introduction belongs to Methods.
Reply: according to the Reviewer's comment we have deleted the last paragraph of the Introduction and we have added it to the Methods section.
Reviewer's comment: line 119: histology is the reference standard, not the index test
Reply: we have deleted the statement in Line 119, according to the reviewer's comment.
Reviewer 2 Report
Castellana et al performed a systematic review of the literature to evaluate the effectiveness of 18F-FDG PET/CT to diagnose malignant vs benign thyroid nodules in thyroid nodules with indeterminate FNA results. They identified 8 original articles and examined results of 18F-FDG PET/CT scans in thyroid nodules with indeterminate FNA results in which there were known histological diagnoses after surgical removal of nodules. Overall, this is a well written manuscript with a detailed review of the relevant data found. My main concern is the confidence with which the authors recommend 18F-FDG PET/CT as a modality to correctly identify malignant vs benign thyroid nodules, as the evidence provided did not clearly show that this is the case. Detailed review and comments below:
Introduction: This is well written and provided a comprehensive summary of the need for a better modality to evaluate thyroid nodules with indeterminate FNA results. It also provided a nice background on 18F-FDG PET/CT scans and the rationale for use.
Methods: This was clearly written and detailed. In section 2.4, it would be beneficial to add more details on the use of DOR as this is the main outcome variable, with some assessment of score results as there is such a wide range of potential score values from zero to infinity.
Results: Also clearly written. In section 3.3, additional details on the papers that excluded certain nodules based on size and TSH would be important to add, as this could lead to significant bias. Also, if all studies include patients who had thyroid nodules removed, this may also lead to patient selection bias as those with higher risk nodules may be the ones included.
Additional explanations in section 3.4 of Figures 3-4 and Table 3 would be beneficial to further summarize the results you are trying to show from these figures.
Discussion:
Paragraph 2-3- While the explanation for some of the false negatives and false positives are helpful to understand, the low sensitivity, NPV, specificity, and PPV remain concerning. It remains unclear based on this discussion how the pooled DOR number can be used instead of the remainder of this data to assess the significance of these findings. In addition, it would be helpful to compare the sensitivity and specificity found here to the other modalities (US, low risk/high risk classification systems) that are being used to assess indeterminate nodules which were discussed in the introduction.
Paragraph 5- Cost is certainly a main concern in using this imaging modality. Additional details in previous studies evaluating cost effectiveness would be helpful.
Conclusions: Again, just unclear how the evidence provided shows that 18F-FDG PET/CT can correctly discriminate between malignant and benign nodules as stated.
Author Response
Reviewer's comment: Introduction: This is well written and provided a comprehensive summary of the need for a better modality to evaluate thyroid nodules with indeterminate FNA results. It also provided a nice background on 18F-FDG PET/CT scans and the rationale for use.
Reply: we are grateful to the Reviewer for having appreciated our systematic review and meta-analysis.
Reviewer's comment: Methods: This was clearly written and detailed. In section 2.4, it would be beneficial to add more details on the use of DOR as this is the main outcome variable, with some assessment of score results as there is such a wide range of potential score values from zero to infinity.
Reply: According to the reviewer's comment we have added more details on the DOR and its interpretation in the Methods section of the revised manuscript.
Reviewer's comment: Results: Also clearly written. In section 3.3, additional details on the papers that excluded certain nodules based on size and TSH would be important to add, as this could lead to significant bias. Also, if all studies include patients who had thyroid nodules removed, this may also lead to patient selection bias as those with higher risk nodules may be the ones included.
Reply: according to the Reviewer's comment, we have added more details on the papers that excluded certain nodules based on size and TSH in Section 3.3. According to the Reviewer's comment we have added a statement about the possible patient selection bias.
Reviewer's comment: Additional explanations in section 3.4 of Figures 3-4 and Table 3 would be beneficial to further summarize the results you are trying to show from these figures.
Reply: We have added additional explanations in section 3.4 of results. We have deleted Figure 3. The previous Figure 4 is now Figure 3 in the revised manuscript.
Reviewer's comment: Discussion: Paragraph 2-3- While the explanation for some of the false negatives and false positives are helpful to understand, the low sensitivity, NPV, specificity, and PPV remain concerning. It remains unclear based on this discussion how the pooled DOR number can be used instead of the remainder of this data to assess the significance of these findings. In addition, it would be helpful to compare the sensitivity and specificity found here to the other modalities (US, low risk/high risk classification systems) that are being used to assess indeterminate nodules which were discussed in the introduction.
Reply: We have added in the discussion of the revised manuscript that the pooled DOR demonstrated that 18F-FDG PET/CT has a moderate ability in discriminating among benign and malignant thyroid nodules with indeterminate FNA. This finding should be considered significant in the current scenario of absence of diagnostic modalities with excellent discriminatory capacity in this setting. A comparison with other modalities is hardly feasible and possibly biased. At least two comparative analyses are available in the literature. Sciacchitano et al., 2017 suffers of methodological issues, including the evaluation of several techniques based only on a limited number of studies (10.18632/oncotarget.17220). de Koster et al., 2018 is described as systematic review, but no method description is reported; also, results are presented only for sensitivity and specificity, while we used DOR as primary outcome since this summary endpoint is not influenced by the prevalence of malignancy, as stated (10.1210/er.2017-00133).
Reviewer's comment: Discussion: Paragraph 5- Cost is certainly a main concern in using this imaging modality. Additional details in previous studies evaluating cost effectiveness would be helpful.
Reply: We have added additional details in the discussion of the revised manuscript about previous studies evaluating the cost-effectiveness of 18F-FDG PET/CT in this setting.
Reviewer's comment: Conclusions: Again, just unclear how the evidence provided shows that 18F-FDG PET/CT can correctly discriminate between malignant and benign nodules as stated.
Reply: we have changed the conclusions reporting that 18F-FDG PET/CT has a moderate ability in discriminating among benign and malignant thyroid nodules with indeterminate FNA.